# Analysis of Socially Vulnerable Communities and Factors Affecting Their Safety and Resilience in Disaster Risk Reduction

Eliška Polcarová *[ID] and Jana Pupíková [ID]

Department of Security and Law, AMBIS University, Lindnerova 1, 180 00 Praha, Czech Republic
* Correspondence: eliska.polcarova@ambis.cz; Tel.: +420-724-328-655

**Abstract:** Human society has been dealing with natural threats since the very beginning of humanity. A society that is better prepared for disasters can better resist the adverse effects of disasters and subsequently adapt to them and thus be prepared in the future for known threats and "new" ones. Level of education, access to information, the income of communities, or social capital are just some factors that can determine the level of safety and preparedness of members of society, especially the vulnerable. For this reason, frameworks and strategies containing disaster risk reduction tools aimed at developing and increasing the level of safety, prevention and preparedness of all states (including island states) for disasters have been created. The article aims to identify vulnerable community members and evaluate the factors that can cause gender inequality in disaster risk reduction and can also significantly influence the increase/decrease of community resilience to disasters. Furthermore, the article presents practical examples from different countries that point to the importance of addressing disaster risk reduction, including global and governmental responses to disasters and the impact of these responses on society.

**Keywords:** community; disaster; resilience; safety; sustainable development goals; vulnerability

## 1. Introduction

As a result of increasing population density and technological development in connection with threats caused by natural forces or human activity, their impacts on human lives, the environment or infrastructure in both developed and developing countries are rising. As countries develop, people become increasingly dependent on services that meet their basic needs. In this case, it is possible to name healthcare, public transport, municipal waste collection or public networks (water, electricity, heat and telecommunications). The failure of these services during or after a disaster has adverse (physical and psychological) effects on the entire population in the threatened area. However, not all countries are at the same level in terms of disaster prevention and preparedness. Based on this fact, the United Nations (UN) decided to unify tools to improve disaster prevention and preparedness. The Sendai Disaster Risk Reduction Framework 2015–2030 [1] is one proactive approach to empowering communities and building their resilience while reducing their vulnerability and meeting the need to prevent, address and eliminate disasters. This framework emphasizes the shared responsibility of all stakeholders and their defined tasks in disaster risk reduction (DRR). However, the primary responsibility for community disaster prevention and preparedness rests with the state. The document aims to suggest means by which to manage disaster risks within and across sectors at all levels and achieve the 17 sustainable development goals (SDGs) by 2030 [2]. Focusing on socially vulnerable groups and the socioeconomic impacts of disasters in disaster risk reduction is also necessary.

## 2. Literature Review

Over the last century, the UN has responded to the increased incidence of disasters [3,4] by issuing strategic documents focusing on reducing risks, mitigating their consequences,

increasing resilience and adapting communities. The first document to assess risk and develop community disaster prevention and preparedness was the *Yokohama Strategy for a Safer World and the Safer World Action Plan: Guidelines for Natural Disaster Prevention, Preparedness and Mitigation* [5]. One of the main priorities of the strategy was to focus on the least developed landlocked and small island developing states. The strategy's main goal was to implement early warning, disaster prevention, preparedness, education and training in the local community. The document emphasizes the community's involvement in all phases of disaster management, especially the empowerment of women and other socially vulnerable groups, in dealing with disasters. The second document, *Hyogo Framework for Action 2005–2015: Building the Resilience of Nations and Communities to Disasters* [6], is based on a review of the previous strategy. New concrete actions have been defined in the areas of community vulnerability reduction, risk assessment and disaster management, including achieving the Millennium development goals (MDGs). The MDGs defined eight international goals to be achieved by 2015. The goals focused on poverty, hunger, maternal and child mortality, communicable diseases, education, gender inequalities, environmental devastation and global partnership. The MDGs have helped nations grow, but the expected level of progress has not been achieved for all indicators in all countries. Therefore, in 2015, the latest strategies, *Sendai Disaster Risk Reduction Framework 2015–2030* [1] and Agenda 2030, *Transforming our world: Agenda for sustainable development 2030* [2], were created, which set out 17 "new" sustainable development goals. These objectives are indivisible and cover the economic, social and environmental fields. The Sendai framework [1] places particular emphasis on the joint responsibility of all stakeholders and their defined tasks in the field of disaster risk reduction. While overall responsibility for disaster prevention, preparedness and risk reduction rests with the state, it is a shared responsibility between governments and relevant stakeholders. Stakeholders include volunteers, community organizations (including women, children and youths, persons with disabilities, the elderly, refugees, or foreigners in general), academia, scientific and research entities, business and professional associations, financial institutions and the media. The document aims to manage disaster risks at all levels (local, national, regional and global) and across these in four priority areas:

- understanding disaster risk;
- strengthening governance to manage disaster risk;
- investing in disaster risk reduction for resilience;
- enhancing disaster preparedness for effective response and to "Build Back Better" in recovery, rehabilitation and reconstruction.

In order to understand disaster risk, it is necessary to know its meaning. For example, UNDRR [7] defines disaster risk as "*the potential loss of life, injury, or destroyed or damaged assets which could occur to a system, society or a community in a specific period of time, determined probabilistically as a function of hazard, exposure, vulnerability and capacity*".

Although most countries have committed to using the Sendai Framework [1] to fulfil the sustainable development goals, inequalities still exist. For example, some community members are not allowed to obtain a quality education (whether based on gender, religion, tradition, or culture), even though this is the basis for fighting poverty and improving health in the community [8].

The Framework for Community Resilience [9], issued by the International Committee of the Red Cross and Red Crescent (IFRC), provides an overview of the approach to community resilience. It primarily focuses on the evolving dynamic nature of communities and the vulnerabilities that threaten communities. Not only does the framework try to define the concept of the community concerning the given issue, but it also defines at what level resilience can be addressed (individual, household, community, local, state, organization, regional and global). Resilience is relevant in all countries because every country has vulnerable communities. The framework, therefore, offers tools and guidance to support community resilience building. Resilience concerns all communities' activities, regardless of discrimination. It is mainly about improving the sustainability and quality of

services and programs. These are provided to the community based on needs, so that they can react and manage disasters in the given area efficiently and in the shortest possible time.

Natural threats do not affect men, women, or children equally. Gender dynamics affect how genders deal with disasters, affecting resilience to existing or new hazards. For example, women with lower socioeconomic status have higher mortality rates than men. Therefore, it is necessary to use such tools and implement such measures to close the gender gap. Based on these facts, the World Bank decided to focus on the issue of gender inequality during disaster management in the document *Gender Dimensions of Disaster Risk and Resilience* [10] for a better understanding of the gender dynamics of disaster risk and resilience. The report assesses how men, women, girls and boys are affected in different ways in disasters. The report aims to identify gender gaps in dealing with disasters, which will be the basis for creating the right policies and programs. If there is a higher need for labor during disasters, this will primarily affect boys. Otherwise, if disasters result in resource constraints, this will affect girls. The impacts of disasters depend on the type and intensity of the hazard, exposure (who and what is at risk), level of vulnerability (susceptibility to damage), preparedness and ability to manage the situation. Gender inequality results from the stereotypes of the position of women and men in society, which are influenced by socioeconomic status (achieved education, position in employment, salary, religion or culture). The level and manner in which women and men prepare and respond to disasters differently affect how women and men recover from disaster impacts. Therefore, the World Bank emphasizes creating disaster risk management tools to mitigate the impacts of disasters while strengthening resilience to close the gap between women and men.

The *Handbook of Community Engagement for Disaster Resilience* [11] emphasizes that community engagement in disaster resolution and disaster risk reduction measures is integral to crisis management. The purpose of this guide is to provide guidance on how to engage communities (including vulnerable members) in disaster response and leverage their strengths (regardless of age, gender, sexual orientation, health status, education, cultural and language capabilities, or socioeconomic status). This manual also provides monitoring, review and evaluation guidelines and community education and engagement guidelines.

## 3. Materials and Methods

For the needs of the given article and understanding of the topic, research, analysis and synthesis of information and data dealing with the issue of resilience, community, vulnerability and disaster risk reduction were carried out. The identification of socially vulnerable community members was supported by research within the dissertation of the first author [12] of this article and is based on a search of selected publications dealing with the issue of vulnerability. The identification of socially vulnerable members of the community was determined based on indicators of social vulnerability: community context and social needs and values. The analysis of factors influencing safety and resilience is based on the sustainable development goals [2] regarding the presence of emergency and rescue services (including their response to disasters) and the existence of appropriate strategies to ensure community safety. These are the main factors that increase the community's safety against disasters, especially at the local level.

### Research and Analysis of the Concept of Resilience and Community

During natural hazards, the resilience and vulnerability of human societies are tested. There are several views on the concept of resilience, depending on the areas represented [9,13]. Arrington et al. [14] generally perceive resilience as "*adaptation despite the risk*". On the one hand, modern foreign definitions perceive resilience "*as the ability of an individual or community to overcome the effects of disasters, or how long individuals or communities can repel the effects of disasters*" [15]. For example, Hopkins [16] applied the concept of resilience to a community and urban context: "*resilience refers to a community's ability not to collapse in the first oil and food shortage and their ability to respond to disturbances adaptively*". For instance,

Montella and Tonelli [17] define this concept as *"the ability of a system to function and develop despite massive shocks and stress"*. According to the UNDRR [7] comprehensive definition, it is *"the ability of a system, community or society exposed to hazards to resist, absorb, accommodate to and recover from the effects of a hazard in a timely and efficient manner, including through preservation"*. In the Czech literature, it is possible to encounter the concept of resilience most often in psychology [18–20], when this expression is most associated with the human ability to cope with adversity and adapt to a new situation. It is the ability to survive, not to break the spirit and not to lose the will to continue living.

The term community comes from the Latin word "communitas" and is often translated as "human community, society, or community". Sometimes, it can also mean "kindness, kindness and companionship," which is associated with giving and accepting mutual service [18]. The community is presented as a group of people living in a defined place connected by joint activities (workplaces, schools and other activities). A community can also be a community of people who share a common interest, profession and religion. Local communities can range from street level to administratively defined borders such as districts, counties, or states. They provide an easily definable area in which the community exists and functions. As a group of people, a community has common relationships based on the community's common interests, such as hobbies, faith, employment, education, sports, politics, or entertainment. Common interests can include skills and resources that the community can use to prepare for disaster response and then use further to reduce disaster risk. A community can also be created based on circumstances in which a group of people is affected by the same event or a common immediate need, such as a terrorist attack, significant industrial accidents, or floods [21,22].

Research and analysis of publications [9,11,21,23] focused on the definition of community and found that a uniform definition of a community is not used in the research area. Therefore, for the research [12] and the article, the following definition was proposed *"a group of people living in a defined territory who share a similar culture, values, customs, norms and resources and at the same time the community is threatened by the same threats"*. The same procedure (search and analysis of publications [24–28]) was also carried out to define the concept of community resilience and the following definition was proposed *"the ability of the community to be self-sufficient and to respond adequately to disasters with the help of available tools, resources and cooperation of stakeholders. Community members' diverse skills and resources can be used to respond effectively, prepare and address the impacts of large-scale local threats and to be able to cope with and adapt to ongoing change"*.

## 4. Vulnerability and Socially Vulnerable Members of the Community

Each affected area has a population group that requires special attention. Historically, socially vulnerable groups have been prioritized in the face of crises or other extraordinary events. For example, "women, old people and children" were rescued from a sinking ship, while "widows and orphans" were given alms. In later times, "unaccompanied children, unprotected women harassed by sexual harassment, the sick and the elderly" were considered the most vulnerable groups. Community resilience is closely related to vulnerability and plays an important role in disaster management (how quickly a community can respond, recover from a negative situation and adapt). Vulnerability thus refers to a state of fragility and disposition to injury. The definition of vulnerability began to be used in the 1970s in the context of risk management to describe the vulnerability of specific communities or territories threatened by environmental or socioeconomic risks, such as earthquakes or food supply disruptions. In the 21st century, the use of the term vulnerability increased significantly with the adoption of the Intergovernmental Panel on Climate Change (IPCC) to assess the potential impacts of global warming at regional and global levels. Based on this fact, vulnerability has come to be used in mitigation and adaptation. Resilience refers to the properties of the system as a whole and vulnerability is focused on the differences between the components of the system. A community may be resilient as a whole, but because it consists of vulnerable population groups, their needs

may not be met by community-level capacities. In this sense, emphasis is placed on the community in which human relationships and interactions are created (for example, the possibility of asking for help from friends, relatives or neighbors) [29–32]. Vulnerability is a set of economic, social and political conditions that differentially affect individual members (or even the entire community) in response to a disaster and the ability to recover from it. However, a high level of vulnerability does not necessarily mean that a community is not resilient; it merely points to the community's inability to resist and respond to disasters. As part of prevention and community preparedness, it is necessary to define the most vulnerable community members living in a defined area threatened by the same disasters. Based on an extensive review and analysis of the literature on vulnerability [15,29–34], this paper defines vulnerability as "*the sum of economic, social and political conditions that differentially affect individual members (or a community) to respond to and recover from a disaster from it*". Disasters disproportionately affect vulnerable groups, especially the poor, ethnic minorities, the elderly and people with disabilities. Vulnerable groups should include individuals who do not have the same opportunities and abilities as the rest of society. Therefore, their vulnerability must be removed or mitigated to match that of others. A community is characterized by a hierarchical arrangement that maintains stability and development that brings change. For example, during disasters, residents face misfortune or pain. It is easier for the residents of this community to cope with the consequences of adverse events than if they were individuals. Helping these vulnerable groups begins with assessing their disaster management needs and resources [18]. Social vulnerability is "*crucially about the characteristics of people and the differential impacts on people of damage to physical structures*". It can be defined as "*the complex set of characteristics that include a person's initial well-being, livelihood and resilience, self-protection, social protection, social and political networks and institutions*" [33]. Generally speaking, social vulnerability describes a community's resilience when external threats threaten it. Natural threats (such as floods, earthquakes and tornadoes), through events caused by human activity or human failure and epidemic diseases, are considered as external threats. Social vulnerability determines the conditions under which the community is susceptible to damage. On the other hand, community resilience tries to deal with danger, recover from it and subsequently adapt. Social vulnerability indicators will help identify community members in critical need of assistance in local hazard prevention and preparedness or disaster recovery. Indicators of social vulnerability will identify these groups: Community context and societal needs and values (Figure 1) are indicators that have helped identify socially vulnerable members of the groups below [29,35].

| **COMMUNITY CONTEXT** | **SOCIAL NEEDS AND VALUES** |
|---|---|
| • demografic distribution of the community<br>• the geografical location of the municipality<br>• the environment<br>• access to the information<br>• community engagement<br>• the infrastructure<br>• experience with disasters | • way of life and life satisfaction<br>• safety<br>• living<br>• food and water<br>• hygienic conditions<br>• social ties<br>• access to the information<br>• livehood<br>• the persistence of ethical and social values |

**Figure 1.** Social vulnerability indicators (modified from [29,35]).

Based on the above factors, the following socially vulnerable community were identified at the local level:

1.   Age vulnerable groups (especially seniors and children);
2.   Children without financial support from their families (orphans);
3.   Cultural and ethnic groups;
4.   Employees of humanitarian organizations and psychosocial intervention teams;
5.   Financially insecure families (primary mothers with children);
6.   Gender inequality (women and girls);
7.   Marginal group (group of people pushed "to the margins of society") Influencing factors: class, ethnic origin, religion, skin color, sexual orientation, gender identity, educational attainment, standard of living, appearance, disabilities, minorities, LGBTQ + people, subcultures, the homeless, immigrants, sex workers, the elderly or young people (ageism);
8.   People (mainly seniors) without family support;
9.   People in collective facilities;
10.   People living in hard-to-reach areas;
11.   People with limitations due to health conditions;
12.   People with low socioeconomic status;
13.   People with severe and persistent mental disorders;
14.   Refugees (escaping due to wars or other armed conflicts, religious, racial or political persecution).

*Gender Inequality in Disaster Risk Reduction*

According to Valdés [36], inequality between men and women (SDG 5, SDG 10) in disaster preparedness and management is a severe problem that needs to be addressed today. The consequences of natural disasters mainly affect the poor and marginalized groups. The reason why poverty affects more women than men is precisely gender inequality, which can be related to the culture, religion, or traditions of the given community. Several factors have contributed to this fact, which is already rooted in the history of human society. The main factor is women's economic disadvantage and financial dependence on fathers, husbands and men. Besides, illiteracy among women and girls is becoming more common. As a result, they are less likely to gain access to safety and disaster preparedness information, which exposes them more to the risks and consequences of these events [37]. Other factors are religious and cultural. Historically, girls do not have the same survival skills as their male siblings.

For illustration, the floods in Bangladesh are mentioned and it is culturally unacceptable for a woman to leave home during the floods. Those women who decided to leave home to save themselves eventually drowned because they could not swim [38]. In their study, Neumayer and Plümper [39] analyzed natural disasters in 141 countries and found that parents prefer sons over daughters when it comes to rescuing their children. The same study also showed that women and girls are more likely to suffer from food insecurity after disasters and do not have safe places with sanitary facilities and a place to sleep. However, due to less security, these women and girls are more prone to sexual assault and rape [40,41].

Today, women and girls are not only at risk from disaster hazards [7] but are also at risk during everyday life due to climate change. Women in most developing countries must provide for their livelihoods (growing basic food and collecting water), with increased weather instability and extreme fluctuations affecting women and girls [34]. For instance, water collection is more time-consuming and physically demanding in desertification areas and women and girls are forced to travel longer distances [37]. In addition, heavy rains and floods cause women and girls to spend much more physical energy and time cleaning the house after floods. The result is less time to provide for households and earnings and less time to study at school and achieve good results [40].

On the other hand, based on their life experiences, these women and girls can better help prepare the community and reduce the vulnerability of the whole community to another natural disaster. Bangladeshi women are an exciting example. These women have decided to take community protection "into their own hands" and adjust the canal basin so as not to cause further flooding in the village in the future [42]. However, the question arises here as to whether men in developing countries are willing to allow women to equalize in today's preparedness and security of the whole community and whether they can accept women's knowledge based on their experience.

The following section presents some statistical data on poverty, education and gender inequalities. The UNICEF report [43] declares that every child has the right to a quality life, regardless of gender and social status in the community. However, many factors can influence children's quality of life-geography, gender, disability, religion, culture and tradition. According to the data presented, more than half of people living in extreme poverty are children (in 2013, an estimated 385 million children lived worldwide on less than USD 1.90 per day). In 2016, a statistical survey [43] was conducted covering 103 countries to determine the number of poor children in households. The result (Figure 2) was that 37% of children and 26% of adults live in poverty worldwide. These data confirm that young people are more often poor than adults. That is why SDG 1 focuses on eliminating extreme poverty among children.

Furthermore, it was also found that children in the poorest 20% of the population are still three times more likely to not live to the age of five than children in the wealthiest fifth of the population. Whether children live to be five years old depends on many factors. For example, it is possible to name newborn complications and infectious diseases (pneumonia, diarrhea, malaria). Another factor is malnutrition, which leads to almost half of child deaths. Since 2000, global under-5 mortality has decreased by almost 47% (from 78 deaths per 1000 live births in 2000 to 41 deaths per 1000 live births in 2016). These facts represent about 50 million children's lives saved. Even so, a very high number of children died only five years ago. For example, in 2016, 5.6 million children died. The goal of SDG 3 is to reduce under-5 mortality, which means, based on available data, that 3 out of 10 countries will need to accelerate their progress towards the poverty and health targets. If the current trends continue as they have been, around 60 million children under the age of 5 could die worldwide by 2030, so meeting the target could save 10 million children's lives [43].

Moreover, more than half of these deaths occur in sub-Saharan Africa and almost a third in South Asia. Despite overall declines in maternal mortality in most developing countries, women in rural areas are three times more likely to die in childbirth than women in cities. Nevertheless, despite global improvements in the social protection system, people with disabilities are still five times more likely to spend vast sums of money on healthcare [43].

Education is the basis for every child's development, social inclusion and poverty reduction. Unfortunately, many children do not have access to quality education and when they do, the range and quality of learning are not always equally guaranteed for all. The goal of SDG 4 strives to ensure that all children complete not only primary education but also further education and achieve adequate academic results. For example, in 1990, only 74 girls were enrolled in primary school for every 100 boys in South Asia. In 2012, the numbers of girls and boys enrolled in the same primary school were equal. Further statistical research from the UNICEF report [43], in which 202 countries participated, shows that as of 2016, 48 countries (about 24%) have met the target and another 25 countries are close to meeting it. Furthermore, 43 countries (21%) should accelerate progress to reach the target on time and the rest of the countries have no data available or insufficient data to present. At least 22 million children in countries may miss out on pre-primary education if rates of progress are not doubled (Figure 3). According to available data, the proportion of children enrolled in pre-primary education is increasing by only 1% per year. Nowadays, only about 2/3 of children in the world are enrolled in preschool education. To achieve the 2030 target, the annual growth rate would have to double to 2.1% per year.

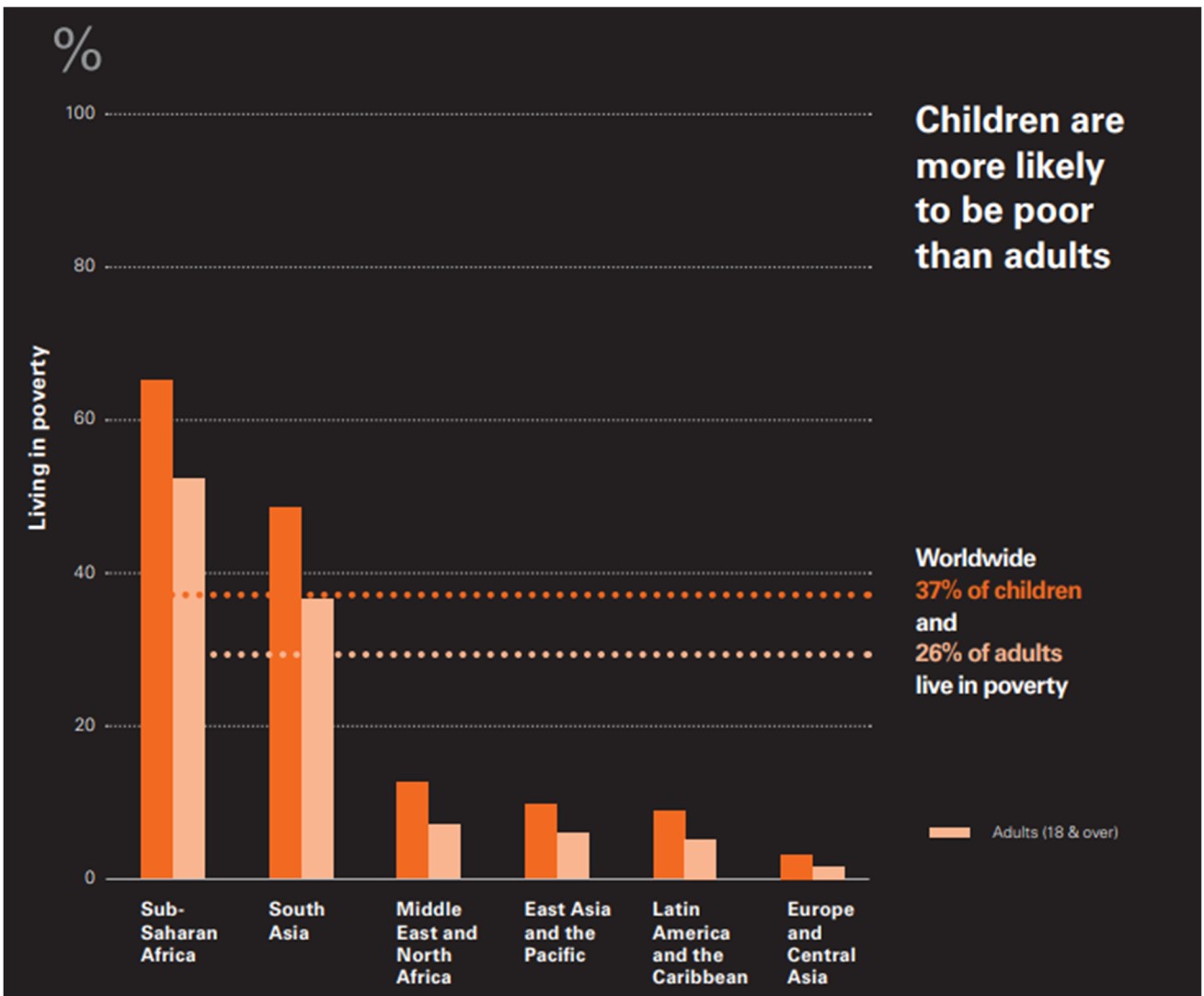

**Figure 2.** Proportion of children and adults living in multidimensional poverty, by region (bars) and worldwide (dotted lines) in 2016 [43].

Another statistic [44] (Figure 4) shows that, between 2000 and 2018, the number of out-of-school girls worldwide decreased from 57 million to 32 million, which is about 44%. The number of boys dropped from 42 million to 27 million in the same period, by approximately 37%. Several barriers prevent children from attending primary school, such as family socioeconomic status, living in seclusion outside infrastructure, armed conflict, insufficient school infrastructure or poor-quality education. High sentiments for school supplies, costs of after-school programs, school meals, approaches to teaching that do not consider the equality of girls and boys, or the cost of transportation, all can be the reason for gender inequality and can thus disadvantage girls in families. These girls may also be at greater risk of early and forced marriage, early pregnancy and health risks during childbirth.

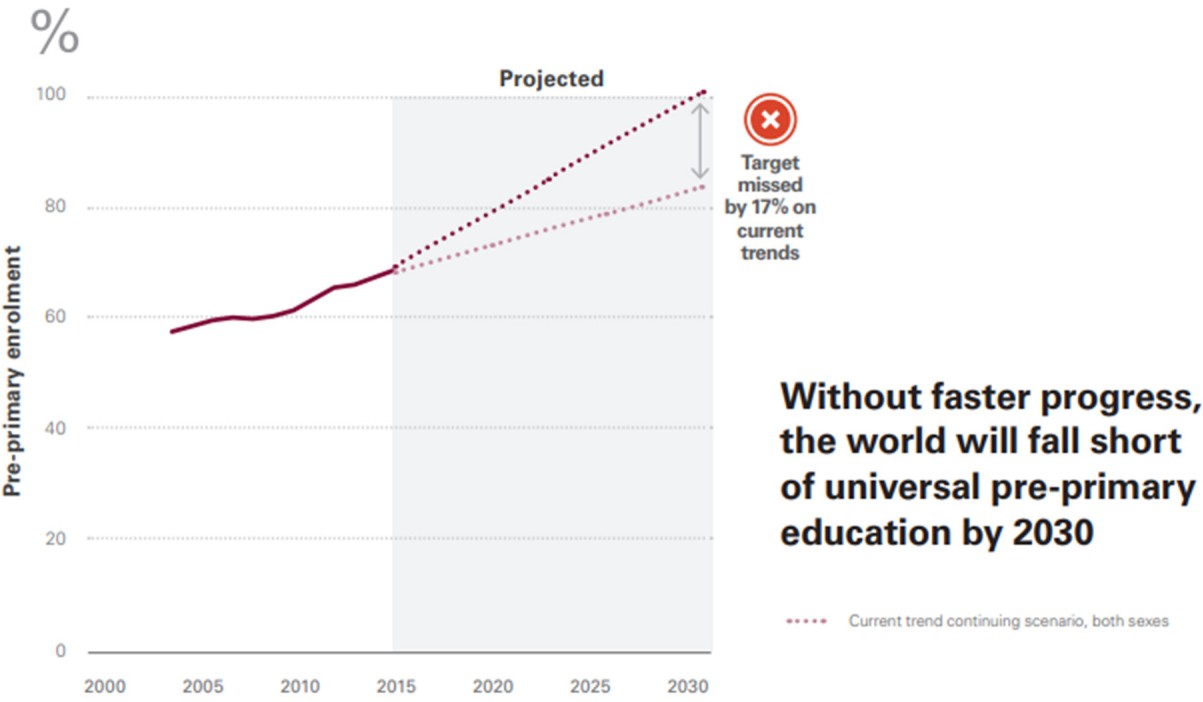

**Figure 3.** Adjusted net pre-primary enrollment ratio (2000–2015) and projections to 2030 for current trends and with acceleration needed to meet SDG target 4 [43].

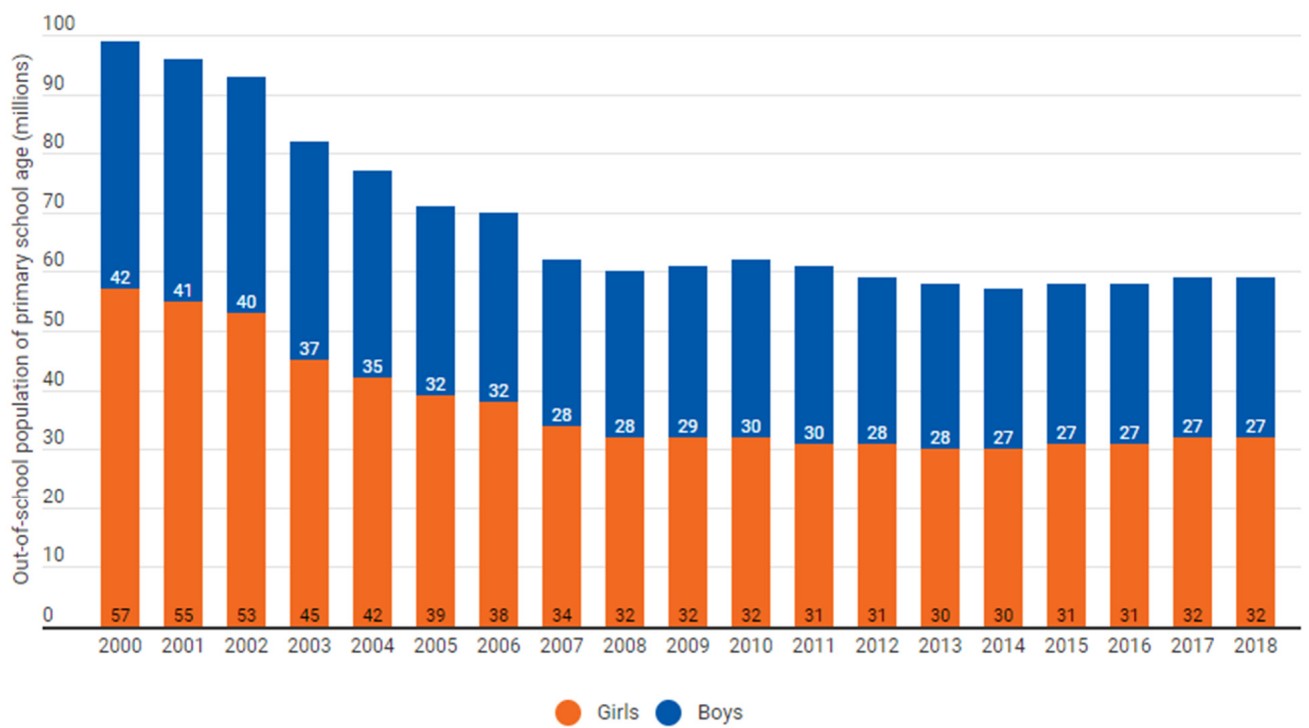

**Figure 4.** Out-of-school population among children of primary school age (millions) by sex (2000–2018) [44].

Unfortunately, gender-based violence is one of today's most widespread human rights violations. According to available statistics from 2021 [45], it is estimated that one in three women (i.e., approximately 736 million women) will encounter some form of violence, either physical or sexual abuse, during their lifetime. This is often from a close person (close family member or partner). In the case of the impacts of the COVID-19 pandemic on society,

the most vulnerable groups (seniors, people with disabilities, children, women, migrants and refugees) have been most affected. In addition, if a woman lives with a partner prone to violent behavior, home quarantine reduces her chances of seeking help by up to 30% in cases of domestic violence against women [45].

## 5. Factors Influencing Resilience and Community Engagement in Disaster Management

The concept of community resilience to disasters focuses on the typical characteristics of a community and individuals and organizations. A resilient community must function well, adapt successfully, be self-sufficient and maintain its social capacity even under stress or shock. A resilient community also has a critical social support system such as neighborhood, family or other kinship networks, social cohesion, interest groups and self-help groups. Communities with higher levels of resilience are better able to withstand disasters and, at the same time, have a better ability to recover from the consequences. In order to properly deal with disasters and eliminate negative impacts, it is necessary to have a fully functional crisis management system. The crisis management cycle consists of four phases of disaster management: prevention, preparedness, response and recovery. Prevention as the first phase of DRR aims to minimize the causes of disasters. The second phase, preparedness, focuses on the preparation of activities during the occurrence of disasters and minimizing the impacts. As part of prevention and preparedness, it is necessary to focus on the availability of resources and means for dealing with disasters and on the education of selected local communities with an emphasis on vulnerable members. It is appropriate to focus not only on threats of a natural character but also on anthropogenic origin. The third phase, response, deals with the response to the resulting disaster. The last phase, recovery, serves to quickly restore essential functions and services in the affected area [46]. The result of the whole cycle should be the acquisition of new experiences, knowledge and lessons learned from past disasters. The acquired knowledge will be the basis for the design of corrective measures and implementation in all phases of the company's crisis management and building resilience in the territory and community [12].

Community resilience is essential in all countries (both developing and developed), as all countries have vulnerable community members. The Sendai Framework [1] points to building citizens' resilience in disaster risk reduction, reducing their vulnerability. Many factors influence the resilience of individuals and communities in disasters. The first factor is access to information (education). Accessible, accurate and reliable information is the basis for disaster preparedness, response and recovery. Nowadays, it is not a problem for citizens of developed countries to find the necessary information on social networks or, at the same time to warn the whole community via mobile devices. Another factor is community income (which is related to education) [47]. Hurricane Katrina can be mentioned as an example. In this catastrophe, average higher-income communities suffered minor damage and recovered much more quickly than lower-income communities. The reason was increased access to resources for preparation, evacuation and disaster recovery [48].

Another factor is social capital. Nakagawa and Shaw [49] define social capital as *"the function of mutual trust, social networks of both individuals and groups, social norms such as obligations and willingness toward mutually beneficial collective action, i.e., pre-and/or post-disaster processes"*. Their article *Social Capital [49]: a missing link to disaster recovery* describes that a local community with substantial social capital was more satisfied with the new urban plan, thus achieving more efficient land use after the earthquake. Against this background, there is a need to invest in "vulnerable" lower-living communities in developing countries to support economic development and stable infrastructure, build financial reserves and thus increase community preparedness and response to disasters.

Several questions arise when developed countries, non-government organizations, the private sector, or other organizations contribute to building community resilience in developing countries (such as Albania, Vanuatu, Nepal, or Indonesia). For example, do the funds provided get to where they are most needed, the "right hands?" Can pressure from

developed countries have a negative impact on the living standards of the community in developing countries? A good idea or a means of help (allowances for orphans, or programs to increase the education of the illiterate) that the wrong people grab can have an adverse effect on the community. International non-profit organizations are interested in supporting children's education, especially orphans in developing countries. However, there is nothing unusual or wrong with this. It is one way to help the community prepare for disasters. What if there are no orphans in the community? Is it possible for the municipality to receive financial support for the education of orphans? What about creating orphans? An example is a Nepalese village [50] that did not have enough orphans to receive financial support from sponsors. Since the communities also live high in the mountains far from civilization, some families decided to "sell" one of the children to the city as orphans because they could not support them. So, the village created orphans "on paper". Yes, these "orphans" will receive primary education within one year, but on the other hand, this situation can have a negative impact. The integration of these children into the community, specifically into their own families, may not be pleasant at all. It is a question of whether the community can accept educated children if they are mainly girls and women. Some cultures (Chinese, Japan) and religions (Jewish, Muslim and Christian) have a clearly defined hierarchy in the family. An educated woman can be the target of psychological humiliation or physical violence. Fulfilling all the sustainable development goals and investing in development, education and raising the standard of living of community members in developing countries is the right thing to do. However, it is always necessary to consider whether helping these vulnerable members will actually help and whether these members will be able to reintegrate into their community and not be excluded on the margins of society.

## 6. Global and Government Challenges in Disaster Management

The impact of disasters affects not only individuals but also entire communities. In addition to physical damage to buildings and infrastructure, there is also an impact on humans' psychological and social aspects. In this case, the public perception of DRR is crucial [51]. According to the disaster phase [52], in the first two phases (Heroic and Honeymoon), people assimilate the initial information about the disaster and shape their "picture" according to the available information on social media. However, taking advantage of this "tragedy" can be an excellent strategy to strengthen the power of governing parties. Klein's idea [53] focuses on the use of disasters and the creation of "own" policies by government officials because citizens are too emotionally and physically distracted to engage, react and effectively resist "illogical" measures.

Currently, the central theme regarding pandemic is COVID-19. Based on the Shock Doctrine [54], it is possible to give an example of the Czech government's response to reducing the spread of coronavirus in the Czech Republic. First, the government banned the sale of respirators to the public. Then, a few days later, the government ordered that people should not move in public without respiratory protection. Finally, the government ordered that a person have respiratory protection during sports, when visiting the forest, or even driving a car (even if alone in the car or with someone in the same household). Failure to comply with these regulations resulted in heavy fines [55].

On the other hand, this reaction of the Czech government showed how the Czech community is resilient and helpful to vulnerable community members. Without waiting for very long, people started buying fabric and sewing cloth masks, which they then distributed free of charge to those who needed them most (seniors, children and the sick). In these cases, it is shown that there is a need to have a sufficiently prepared community for existing threats and new ones.

Another example is the Hurricane Katrina disaster, which caused enormous damage. Shughart [56] describes massive failures in local, state and federal government. Government officials were given early warning that a hurricane was approaching and had plenty of time to prepare, but they failed to act on the warning. The first significant failure was confusion

over information to the public that was flawed or inaccurate, while key officials were not adequately trained in their roles. Another reason for failure was a lack of preparation. The government was not prepared even though meteorologists predicted the likelihood of a hurricane. A year before Hurricane Katrina, a simulation exercise was conducted to identify gaps in hurricane preparedness and implement appropriate measures in plans. Even though there were flaws, the government failed to learn from this simulation, which became evident a year later during Hurricane Katrina. Here it is possible to mention the complete breakdown of communication that overwhelmed the key actors who could not communicate with each other. The indecisiveness of government officials regarding the deployment of supplies, failure of supplies, medical personnel and the assumption of responsibility for shortages resulted in a lengthy resolution and reconstruction of the territory after the disaster. Fraud and abuse were the final factors that slowed down the territory's recovery [57].

The state's "negative" endeavor can be a significant problem in implementing the Sendai framework [1] principles in all countries. In contrast, the framework emphasizes that the safety of its citizens is the state's responsibility. Assessing the impact of disasters on the community is essential to reducing the community's vulnerability. Unfortunately, estimating these impacts on disasters before the disaster itself and assessing the potential consequences is a complex process that needs time and money to analyze the impacts [58]. Here the question arises as to whether the government has an reason to address this lengthy, costly process, even if the disaster may not occur under their rule.

## 7. Results and Discussion

According to the Universal Declaration of Human Rights and several international conventions dealing with human rights, including Act No. 2 Coll., *In the Charter of Fundamental Rights and Freedoms*, every person has the right to life, freedom of speech, quality health care and quality education without discrimination. Unfortunately, not every individual has the opportunity to exercise these rights. This fact can be seen when dealing with disasters in developing countries for women and children and the female gender is marginalized mainly for religious and cultural reasons. In 2015, a total of 17 sustainable development goals were created, which would help to achieve these fundamental human rights in all countries of the world, especially for vulnerable people such as women, children and other physically and psychologically weaker individuals (Section 4). However, based on the literature review (Section 2) and analysis of selected examples from practice and statistical data, it is necessary to state that, despite the existing strategies and documents dealing with DRR and SDGs, these concepts are not implemented and observed everywhere. At the same time, it is necessary to state that, in order to solve disasters and eliminate negative impacts correctly, it is crucial to have a fully functional crisis management system.

Furthermore, there is a need to implement approaches and tools that ensure disaster risk reduction (phases of prevention, mitigation and preparedness), building community resilience and reducing community vulnerability. At the end of each cycle, it is necessary to assess the situation and implement the recommendations in the "news" cycle. After this phase, a shift in disaster risk reduction and building the resilience of the entire society should be observed in the selected territory (Section 5). Nevertheless, despite advanced technology, it is still difficult to accurately predict the occurrence of any adverse event these days. Considering that the human population is growing increasingly and there are higher demands on the standard of living, it is necessary to implement such strategies and tools, based on which the community will be able to respond adequately to the ongoing situation with minimal impacts.

## 8. Conclusions

In conclusion, women and men often have different but always impressive views on adapting to negative influences. However, gender equality must be considered when making decisions [59–61]. From a theoretical point of view, it is necessary to increase the

possibility of educating women and girls in developing countries such as Bangladesh or Nepal and thus increase their participation in decision-making in communities within the framework of DRR. From a practical point of view, it is not that simple. The reason may be a cultural or religious factor that stems from the overall history of the community. The question is whether the men in these communities are willing to allow women to match them today in ensuring preparedness and reducing the vulnerability of the entire community and, also, whether men can accept the knowledge of women based on their practical experience. Strategic tools aimed at the community try to help strengthen their resilience and create means for prevention and preparation of individual members for various future or new dangers. When implementing these tools, it is necessary to focus on people with special needs, that is, on vulnerable members of the group and to design an appropriate strategy to help them cope with disasters like other members can. The Sendai Framework [1] helps ensure this involvement ensuring the equality of all members in dealing with disasters, especially in the case of natural hazards. In addition, this framework emphasizes that (international) cooperation (SDG 17) and the participation of all stakeholders are necessary for prevention and preparedness in DRR. Although the Sendai Framework is, from a historical perspective, an excellent start for some communities to promote the equality of individual (vulnerable) community members regardless of gender, age, culture, religion and geographic location, for some communities, it can be a disruption to their way of life. Religious and cultural factors can be key in finding solutions to these problems in increasing community resilience and thus fundamentally disrupting the course of DRR in all disasters.

A limitation of the research is that it is difficult to "push" states to implement DRR issues in their national strategies for the local community. Each state is specific with regards to demographic location, political situation, religion, customs and traditions, laws and institutions. Therefore, there is a need for the solution of disasters and DRR to be solved mainly at the local level, focusing on the weak and on different community members.

**Author Contributions:** Writing—original draft, E.P.; Writing—review & editing, J.P. All authors have read and agreed to the published version of the manuscript.

**Funding:** This research received no external funding.

**Institutional Review Board Statement:** Not applicable.

**Informed Consent Statement:** Not applicable.

**Data Availability Statement:** Not applicable.

**Acknowledgments:** The authors would like to thank AMBIS University, Prague for its support.

**Conflicts of Interest:** The authors declare no conflict of interest.

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
