# Peer review of "Analysis of Socially Vulnerable Communities and Factors Affecting Their Safety and Resilience in Disaster Risk Reduction"

_sustainability, doi:10.3390/su141811380_

Round 1
Reviewer 1 Report
The study reflects an understanding on examining socially vulnerable communities and factors affecting their resilience in disaster risk reduction. However, some improvements are needed to make the paper suits the journal's standard. Detailed comments are in the document.

Author Response
Dear Reviewer,
Thanks for all your suggestions to improve the article. We have reviewed all your suggestions and incorporated them into the article. More information is in the attached document.
Sincerely,
Authors.

Reviewer 2 Report
Please refer to scientific sources that published in 2021 and 2022.

Author Response

(The authors gave the same response as above.)

Reviewer 3 Report
1)The author should include more details about the study area with relevant research in the introduction section.
2)The author should give a brief overview of the main disasters and their related socioeconomic effects in the Czech Republic.
3)The author needs to increase Figure 1's image resolution.
4)On page-3, The Colorado Resiliency Framework's significance should be discussed by the author.
5)The author needs to explain the connection between social vulnerability and a Community Resilience to withstand hazards.
6)Author should include the methodology framework graphically for this study for better understanding the flow of research work.
7)The terms "vulnerability,” “Social Vulnerability” and “Disaster Risk," should be defined by the author.
Author Response

(The authors gave the same response as above.)

Reviewer 4 Report
The paper titled "Analysis of socially vulnerable communities and factors affecting their safety and resilience in disaster risk reduction" (sustainability-1859709) has been an interesting read because of its particularly relevant topic. Therefore, I support this idea and think that it might become a valid contribution to the literature so far.
What I miss is a clearer structure. Moreover, I am not "bothered" by the absence of models etc. - in the contrary, given its over-use, I feel that this might be a strength - but by the lack of a clear focus. For instance, the authors mention several examples (e.g., from Hurricane Katrina or a Nepalese village) without deepening too much. In my opinion, the authors either have to strengthen/amplify each example they provide or they have to reduce the number of examples while focussing on a more selected (i.e., limited) number of them. Moreover, when they mention "some statistics" from the UN I suggest to create a couple of tables/graphs with historical data series just to show the evolution of these statistical items over time.
This is an innovative and promising paper, but the authors should review each sentence of it and ask themselves "what could I add?" or "how could I improve this argument?". No big, radical changes required but a very careful, sentence-by-sentence review.
Author Response

(The authors gave the same response as above.)

Round 2
Reviewer 4 Report
Thanks for your amendments, which are fine for me and significantly improve the quality of the manuscript.